# Estimation of Spatio-Temporal Parameters of Gait and Posture of Visually Impaired People Using Wearable Sensors [note 1]

**DOI:** 10.3390/s23125564

**Published:** 2023-06-14

**Authors:** Karla Miriam Reyes Leiva, Miguel Ángel Cuba Gato, José Javier Serrano Olmedo

**Affiliations:** 1Engineering Faculty, Universidad Tecnológica Centroamericana UNITEC, San Pedro Sula 21101, Honduras; karla.reyes@ctb.upm.es; 2Center for Biomedical Technology (CTB), Universidad Politécnica de Madrid, 28040 Madrid, Spain; ma.cuba@alumnos.upm.es

**Keywords:** inertial sensors, O&M, postural assessment, rehabilitation

## Abstract

In rehabilitating orientation and mobility (O&M) for visually impaired people (VIP), the measurement of spatio-temporal gait and postural parameters is of specific interest for rehabilitators to assess performance and improvements in independent mobility. In the current practice of rehabilitation worldwide, this assessment is carried out in people with estimates made visually. The objective of this research was to propose a simple architecture based on the use of wearable inertial sensors for quantitative estimation of distance traveled, step detection, gait velocity, step length and postural stability. These parameters were calculated using absolute orientation angles. Two different sensing architectures were tested for gait according to a selected biomechanical model. The validation tests included five different walking tasks. There were nine visually impaired volunteers in real-time acquisitions, where the volunteers walked indoor and outdoor distances at different gait velocities in their residences. The ground truth gait characteristics of the volunteers in five walking tasks and an assessment of the natural posture during the walking tasks are also presented in this article. One of the proposed methods was selected for presenting the lowest absolute error of the calculated parameters in all of the traveling experimentations: 45 walking tasks between 7 and 45 m representing a total of 1039 m walked and 2068 steps; the step length measurement was 4.6 ± 6.7 cm with a mean of 56 cm (11.59 Std) and 1.5 ± 1.6 relative error in step count, which compromised the distance traveled and gait velocity measurements, presenting an absolute error of 1.78 ± 1.80 m and 7.1 ± 7.2 cm/s, respectively. The results suggest that the proposed method and its architecture could be used as a tool for assistive technology designed for O&M training to assess gait parameters and/or navigation, and that a sensor placed in the dorsal area is sufficient to detect noticeable postural changes that compromise heading, inclinations and balancing in walking tasks.

## 1. Introduction

Gait analysis plays a vital role in various health applications, but the use of optical motion analysis systems for measuring spatio-temporal gait parameters has limitations such as cost, fragility, lack of portability and resource requirements [1]. Inertial measurement unit (IMU) sensors have emerged as an alternative method for measuring gait parameters, capable of providing both gait and posture measurements by combining IMUs with magnetometers [2].

Among the populations that could benefit from gait measurement, visually impaired people (VIP) stand out. Although electronic navigation aids have been developed for VIP, many of them rely on complex architectures that pose challenges for environmental sensing [3,4]. Smartphone-based inertial sensing, utilizing deep learning methods, requires extensive data for training and may not be tailored specifically to the unique gait patterns of VIP [5]. Furthermore, a recent review highlighted the lack of inertial sensor systems designed for VIP and the scarcity of literature on IMU-based biomechanical analysis in VIP-oriented applications [6]. While wearable inertial sensors have gained attention in clinical research for the gait parameters of people with conditions such as stroke, Parkinson’s and multiple sclerosis, there is currently a dearth of studies focusing on VIP biomechanics [7,8]. Existing non-wearable systems for VIP gait analysis have limitations, and there are a lack of user-oriented wearable systems designed specifically for this purpose [1]. Moreover, most spatio-temporal gait parameter analyses for VIP have been developed using motion tracking systems, and the assessment of independent mobility and rehabilitation in orientation and mobility (O&M) typically relies on visual estimates rather than quantitative measurements [9].

Considering the aforementioned gaps, this research aims to propose a simple architecture based on wearable inertial sensors for quantitatively estimating gait and postural parameters in VIP. By utilizing IMUs and magnetometers, this approach seeks to provide accurate and user-friendly gait assessments tailored to the needs of visually impaired individuals, facilitating their orientation and mobility rehabilitation. The study design includes validation tests with visually impaired volunteers performing various walking tasks, with a focus on assessing gait characteristics and natural posture. By addressing the limitations of current assessment methods and offering a practical solution, this research contributes to the development of assistive technologies for VIP and has the potential to enhance their independent mobility and navigation.

This article presents two IMU-based methods for measuring step length (SL), gait velocity (GV), step count (SC) and total displacement (D), and one method for postural assessment (PA). The methods are novel due to their combination of several factors. First, the methods use absolute orientation angle values (leg rotation) during the gait cycle to calculate SL and GV, rather than relying on accelerometry or angular velocity values. Additionally, they apply a gait biomechanical model instead of an abstraction model or a direct integration model to avoid accumulated drift error when calculating each SL and displacement, and this approach maintains the simplicity of the system setup with low computational processing [10]. Second, the methods provide precise real-time measurement of spatio-temporal gait parameters despite using only one or two low-cost wearable inertial sensors. This eliminates the need for post-acquisition gait analysis software to calculate parameters and process data in clinical approaches, as seen in [9]. Third, the methods are designed to be user-oriented and implemented as assistive devices for O&M training, with specific conditions set in the measurement algorithms.

## 2. Materials and Methods

A measuring system comprising two BNO055 (Bosch Sensortec Reutlingen, Kusterdingen, Germany) 9DOF inertial sensors was previously developed [9]. The postural assessment was conducted using a MetaMotionR (MBIENTLAB, Inc., San Francisco, CA, USA) IMU.

### 2.1. Biomechanical Principles

In a gait cycle formed of two consecutive steps, θ (Figure 1A) refers to the inclination angle with respect to the vertical angle, as measured by the inertial sensor. For its part, γ is the rotation angle between the heel strike and heel off during the stance phase of the gait cycle of both legs. However, as the knee does not remain stiff during the gait, γ is, more accurately, the rotation of the virtual segment connecting the hip with the heel. To determine this segment, the inclination, and the length of both of the main subsegments of the leg (thigh and shank), must be known; this adjusted model was the biomechanical model used for the estimation in both of the methods, and it is shown in Figure 1B.

In the two-sensor (TWS) configuration, both inclinations were measured using the IMUs. In the one-sensor configuration (named “thigh sensor” TS), the sagittal flexion–extension angles of the knee at heel strike and heel off were extracted from the literature [11,12], and the unknown inclination was calculated relying on the fact that α = ∥θ′ − θ″∥, where θ′ is the thigh inclination, θ″ is the shank inclination and α is the flexion–extension angle. The rotations denoted by γ led to the displacement of the center of mass in the *x*-axis of the navigation coordinate frames, which matched the direction of the current step (k). Thus, this displacement is a more accurate approximation of the step length.

### 2.2. Experimental Procedure

This study employed a single-group design with a mixed methods approach. The study was approved by the ethics committee of the Universidad Politécnica de Madrid in October 2020 (ref. ID: 2020000224) and was conducted following the principles of the Declaration of Helsinki. To test both of the methods, nine visually impaired volunteers were recruited via e-mail using the collaboration pool of the National Spanish Blind Association and word of mouth, and the sample selection criteria involved visually impaired people that use either long cane or guide dog assistance. All of the sessions were video recorded, and the volunteers provided their consent after the experimenters read the consent form to them. All of the experiments were conducted at the subjects’ residences. The volunteers were instructed to walk in five different conditions with varying velocities, including four indoor experiments and one outdoor experiment. The total displacement for each subject depended on the dimensions of their residence and included the following conditions: Exp1—walking 7–10 m at a normal velocity from point A to point B, Exp2—walking 7–10 m at a fast velocity from point A to point B, Exp3—walking 7–10 m at a slow velocity with short strides from point A to point B, Exp4—walking 14–20 m at a normal velocity with two stops in a hallway from point A to point B and Exp5—walking 60–100 m outdoors from point A to point B. The data collection comprised a total of 2068 steps analyzed over 1039 m to evaluate the accuracy of the two estimation methods.

### 2.3. Parameter Measurement

A single SC algorithm was utilized for both the TWS and THS methods. The algorithm was designed to identify a new step when the leg crossed the vertical position, and to detect the local maximum and local minimum in each inflection point of the rotation angle patterns. Additionally, a straightforward activity recognition algorithm, based on angular velocity, was developed to determine whether the user was moving or not, and thus to prevent the detection of “false steps”. More conditions were incorporated, requiring each local maximum to be followed by a local minimum (and vice versa) to count a new step.

Regarding SL, the study used the calculated mean accepted value to obtain the correct mean absolute error, along with the registered SC by the volunteers and the defined total displacement. The value of SL was obtained from the rotation *γ* using the following equation, which applies the cosine law:$${SL}_{k}=\sqrt{2\times {hh}_{i}^{2}-2\times {hh}_{i+1}^{2}\times cos\;cos\;\gamma_{k}}$$

The length of step *k*, denoted by *SL_k_*, was calculated using the cosine law by considering the lengths of the hip–heel segments for each leg, denoted by *hh_i_*.

The distance measure D refers to the summation of the measured step length for every identified step, as illustrated in Figure 2. This implies that any absolute error in the SL and SC measurements directly influences the D value. The GV for each method was determined by dividing D by the total walking time. The walking time was calculated using the step detection algorithm and compared to the estimated value from the video recordings for computing the mean absolute error, thus ensuring accurate error in GV. The SL and SC measurements can also affect the velocity error.

Postural stability refers to the body’s ability to maintain balance, and it is often evaluated via postural sway. Postural sway involves constant adjustments of the body’s center of gravity on a relatively narrow support base, and personal visual feedback plays a significant role in controlling balance [13]. In this study, postural assessment (PA) was conducted using orientation angles obtained from a sensor placed on the subjects’ backs to provide input under dynamic conditions. The approach utilized in this study adheres to the principles of posture monitoring, where a single sensor is positioned at the end of the cervical curve and the beginning of the thoracic curve, as demonstrated in [14]. The 3-angle orientation sensor enables the detection of bending/inclinations, subject tilting to the left and right and postural heading. Although a single method was employed to analyze postural balance, this assessment was performed at a deferred time compared to the other real-time methods that were tested. The standard deviation of the measured roll angle, which indicates the variation in left and right tilting, as well as the standard deviation of the pitch angle, which represents the variation in inclination, were considered in the analysis. In order to confirm that the standard deviations obtained from the orientation angles were capable of detecting significant postural changes that affected heading, inclinations and balance during walking tasks, the researchers conducted a posture evaluation based on the video recordings, as described in [15].

## 3. Results

### 3.1. Subject Information and Gait Characterization

Nine volunteers ranged in age from 23 to 70 years old (M = 51.6, SD = 14.9) The volunteers had been blind for an average of 20.6 years (SD = 14.2) (Table 1). Three participants had residual eyesight (or only bright light perception), self-reported as the recognition of 5–10% shapes. All of the volunteers stated that they could move around Madrid independently with a long cane, and five also had used a guide dog for an average of 5.2 years (4.6 SD). The volunteers had used white canes for an average of 17 years (SD 14.88) and most (8/9) had had at least one O&M training session. Among them, six volunteers had no remaining vision. The gait characteristics of the volunteers were considered as the ground truth (GT) values for the estimation of accuracy during the validation of the methods (Table 2). The values were obtained by measuring the walked distance and counting the steps walked. This assessment was conducted visually and validated via video recording. The average step count from all of the experimental conditions was two steps per meter, with the highest standard deviation observed in the high-velocity (11.36) and outdoor (11.52) conditions.

### 3.2. Method Evaluation

#### 3.2.1. Step Count and Step Length

The step count algorithm demonstrated that even with a complete stop (i.e., no rotation angles during several seconds) and slight variations in the heel strike pattern, the step count algorithms worked with precision, resulting in a mean error of 1.48 ± 1.56.

The mean absolute error of the measured SL for Experiments 1, 2, 3, 4 and 5, for all nine subjects, was 4.59 ± 3.69 cm for the TWS method and 7.37 ± 4.68 cm for the THS method. Table 3 provides a summary of the mean error for TWS and THS, considering different velocities. The TWS method presents a lower error in all of the experimental conditions. However, THS presents an error lower than 10 cm in all of the experimental conditions (Table 3).

#### 3.2.2. Distance and Gait Velocity

Figure 2 is a representation of the measured distance according to SL and SC of three random subjects with different gait velocities. As can be observed in the graphic and the error results, small stepping and slow velocity might have less accuracy for both methods.

The overall mean absolute error for D for all nine subjects in the 1039 m walked was 1.78 ± 1.80 m for the TWS method and 3.20 ± 3.38 m for the THS method. It should be noted that there is a variation in accuracy in the mean values while changing indoor/outdoor conditions; there is an increase in the absolute error for THS from 2.03 ± 1.92 to 7.9 ± 2.74 m, while the TWS error remains nearly unchanged (~1 m).

In the case of GV measurements, there is negligible variation in error between the THS and TWS methods across all of the experimental conditions. The mean absolute value was 10.40 ± 6.88 cm/s for TWS and 7.03 ± 7.13 cm/s for THS, as both methods employed the same approach for assessing time during gait cycles. Figure 3 depicts the measurement values for two randomly selected subjects across all of the experimental conditions. Figure 3 illustrates how both methods can detect changes in velocity for Experiments 3 and 4, and that the normal velocity measured for Experiments 1, 2, and 3 using the TWS method is more accurate. Overall, the TWS method exhibited higher accuracy but THS presented a lower SD across all experimental conditions and for all spatio-temporal gait parameters measured.

### 3.3. Postural Assessment

Regarding the assessed postural stability, on the one hand, researchers observed noticeable tilts and inclinations during the walking tasks and classified them as binary options (Y/N). On the other hand, they calculated the average standard deviation (SD) of the measured postural angles for each subject and compared it to a threshold value, which was an estimate of a normal angle deviation based on the acquired data from the subject sample. Values exceeding the threshold were categorized as Y and the rest were labeled as N. Based on the analysis of the 45 walking tasks, the % error of the estimation was 38%.

## 4. Discussion

This study aimed to use low-cost sensors and to compare two algorithms using data from IMUs placed on the leg and upper back to estimate parameters related to gait and posture that are typically visually assessed by rehabilitators in O&M rehabilitation of visually impaired individuals. The ultimate objective was to select a method that provided the more accurate estimations, which could allow for proximate quantification of the information that was assessed without quantification methods in the current examination methods during O&M.

According to the literature, the proposed methods that use inertial sensors placed on the leg segments concentrate on two approaches for determining gait parameters. One approach involves identifying related variables that correlate with the unknown parameters. Another approach involves direct estimation using biomechanical models or kinematic chains. Frequently, stride length is calculated first by estimating the distance between the feet. The inverted pendulum model, which uses changes in the center of mass height, can also be used. Alternatively, the horizontal acceleration of the feet can be double-integrated to calculate stride length [16]. The approach proposed in this study aims to utilize leg rotation angles with leg segmentation. While the method has shown reasonable accuracy, a limitation is that the method heavily relies on the correct positioning of the sensor on the leg, and as it is based on the orientation angles of the leg during the walking stages in the gait cycle, a bad positioning of the sensor could jeopardize the accuracy of calculations. Nonetheless, depending on leg rotations, this can also be advantageous in terms of detecting steps taken and rest periods more accurately, which can provide better insight for navigation tools. Some researchers have developed accurate methods of detecting steps using acceleration and rotation values in order to reconstruct paths for indoor navigation using deep learning models such as the uni-directional long short-term memory (LSTM) model [17]. This method for real-time assessment could be beneficial for the development of a more complex O&M rehabilitation tool. 

A large percentage of the visually impaired population is elderly and their gait characteristics may differ from those of non-visually impaired adults. Recent studies have shown that non-visually impaired adults have a mean gait velocity ranging from 1.39 to 1.49 m/s [9], while blind or visually impaired individuals exhibit a mean gait velocity between 0.86 and 1.11 m/s [18]. Lower velocity values would likely have bigger standard deviations. Our approach is an acceptable method to measure velocity; however, the error must be reduced if greater accuracy is required in a rehabilitation examination. According to Lim et al. [19], most of the current literature methods focus primarily on other impairments, such as mobility impairments, or different clinical scenarios; none are yet focused on developing methods to analyze gait velocity in visually impaired populations. 

The results of this study suggest that an automated method for the real-time detection of postural changes to assist in making postural assessments could be achieved with the proposed method. However, it is recommended that the study be expanded to include a larger group of participants to determine a threshold value for different gait velocities; this could result in a lower degree of error. 

Finally, by utilizing the input data from the system, including SC, SL, GV, D and PA, it is possible to estimate additional spatio-temporal parameters that may be relevant to O&M rehabilitation, such as cadence or stride rate, which measure the number of steps or double stance per minute [20]. For instance, by applying the same condition used to determine whether a person is walking or not for SC, it is possible to measure the time a user pauses between steps when moving from point A to point B during a displacement task. The results suggest that a balance between general upper body and gait assessment can be achieved with only 2–3 inertial sensors, whereas the current literature may suggest using a greater number of sensors [21]. The results indicate that both the TWS and THS methods could be employed to develop tools for O&M training. However, the TWS method is more appropriate due to its greater accuracy. A rehabilitated gait that promotes enhanced stability can improve balance, emphasizing the importance of early initiation of orientation and mobility (O&M) training to enhance gait, balance and movement [1]. The gait patterns of visually impaired individuals differ from those without visual impairment. Therefore, assistive technology development and validation should involve insights from visually impaired individuals. The insufficient involvement of end-users is a limitation of the current literature; development processes must consider targeted users to ensure that usability and accuracy are not compromised [6]. The dataset created in this research could also be beneficial as an input for deep learning models that train quantitative parameters of walking, as evidenced by a recent review [18]; there is a need for research into gait in free-living conditions, i.e., at end-user residences.

## 5. Conclusions

Two different methods were created and tested in five different experimental conditions to assess gait spatio-temporal parameters and to test which method could be more accurate. The results from the TWS configuration were more precise for D, SL, and GV, while the THS configuration produced comparable but slightly less accurate results with greater SD and error, with differences of ±1.42 m absolute error for D, ±2.78 m error for SL, and ±3.37 cm/s error for GV. The addition of a second wearable sensor increases both the computational and hardware cost of TWS, which is the significant architectural difference between the gait estimation methods. Therefore, for measuring the O&M rehabilitation parameters of gait, the TWS method could be more appropriate. However, these results suggest that the THS method, which uses only one sensor, could also be used for other applications in assistive technology when this accuracy range is acceptable. After validating a method for measuring the parameters accurately, the subsequent steps in this research would be to consider developing a feedback system for the user and O&M specialist, as well as the development of a digital platform that considers the end-user’s usability as an assistive technology designed for O&M training. This mixed methods approach aimed to validate the proposed method and architecture for O&M training and assess their potential as assistive technologies for visually impaired individuals.

## Figures and Tables

**Figure 1 sensors-23-05564-f001:**
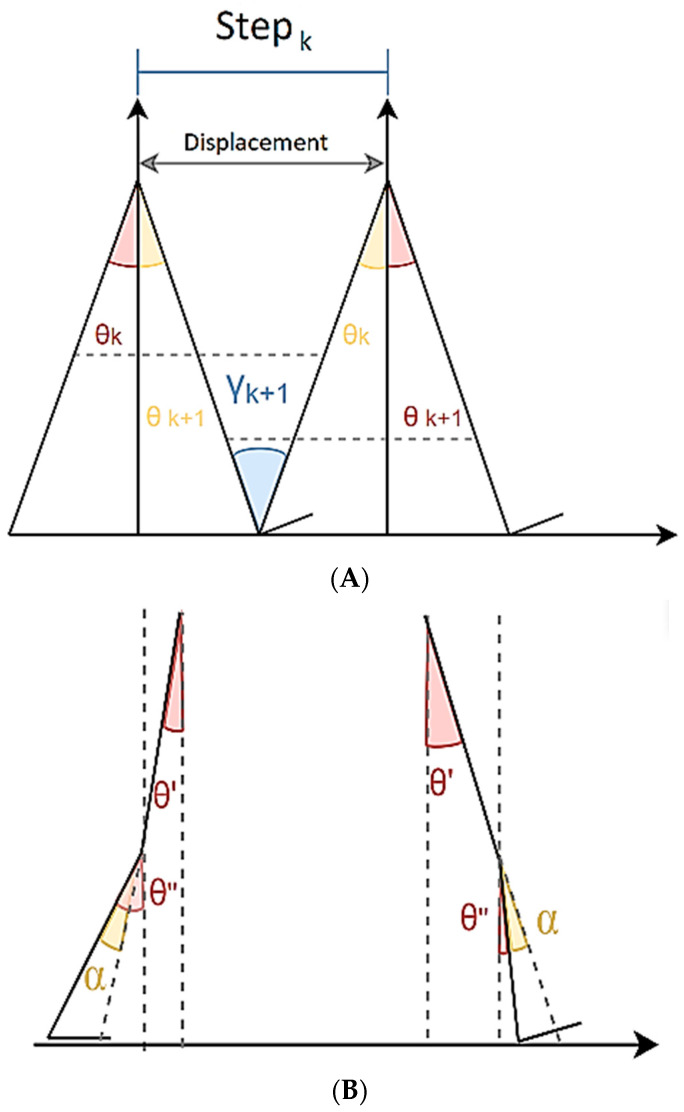
Biomechanical models: (**A**) basic model, (**B**) knee flexion–extension adjustment.

**Figure 2 sensors-23-05564-f002:**
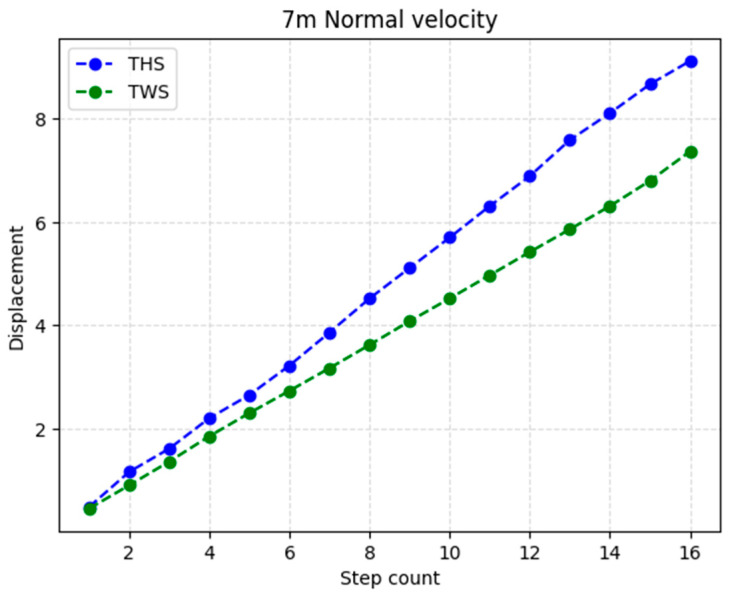
Step length measurements in each step count for Exp2 in S3 and S4.

**Figure 3 sensors-23-05564-f003:**
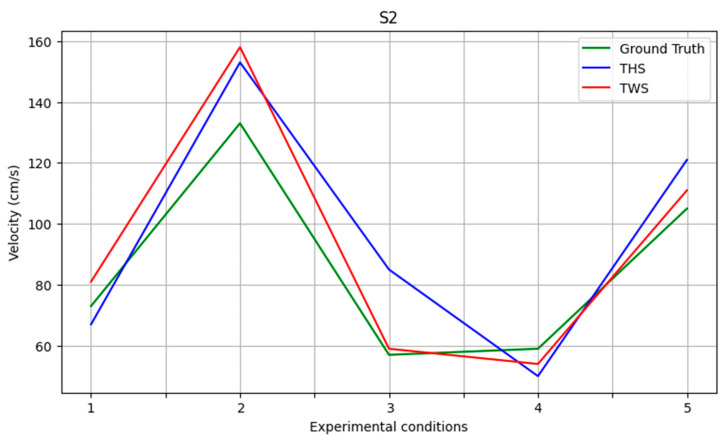
Gait velocity measurements in two random subjects for all experimental conditions.

**Table 1 sensors-23-05564-t001:** Socio-demographic characteristics and information concerning the visual impairment of participants.

Age	Gender	Condition	ROV	YOB	RHB	YWGD	YWC	O&M Training
23	M	Optic nerve atrophy	No	11	Yes	1	10	2 weeks
31	M	Retinoblastoma of the optic nerve	No	28	Yes	N/A	17	FS
27	M	Glaucoma	No	6	Yes	2	5	1 day
24	F	Congenital malformation	No	24	Yes	8	19	FS
26	F	Leber’s congenital amaurosis	5%	26	Yes	13	16	1 year
23	F	Retinitis pigmentosa	5%	7	Yes	N/A	4	3 months
70	M	Myopia	No	52	No	N/A	55	No
23	F	Bilateral congenital malformation	No	6	Yes	N/A	6	3 months
39	M	Retinitis pigmentosa	10%	26	Yes	2	25	2 months

ROV = rest of vision, YOB = years of blindness, RHB = rehabilitation, YWGDs = years with guide dog, YWCs = years with cane, FSs = few sessions.

**Table 2 sensors-23-05564-t002:** Step length ground truth values.

	S1 (cm)	S2 (cm)	S3 (cm)	S4 (cm)	S5 (cm)	S6 (cm)	S7 (cm)	S8 (cm)	S9 (cm)	Average	SD
Normal velocity	57.2	57.4	43	41.3	56	48.8	37	52.3	61.6	50.48	7.95
High velocity	66.7	66.7	50	38.9	69	50	43	60	71.4	57.37	11.36
Low velocity	55.6	40	33	41.2	47	45.5	34	56.3	50	44.81	7.91
Outdoor normal velocity	76.9	56.6	41	48.9	51	57.1	38	48.4	66.7	53.88	11.52
Step count/m	1.62	1.85	2.68	2.38	1.71	2.01	2.65	1.87	1.63	2.04	0.40

**Table 3 sensors-23-05564-t003:** Mean absolute error with standard deviation for step length measurements in TWSTHS.

	Normal Velocity	Low Velocity	Fast Velocity	All Velocities	Indoor	Outdoor
THS	8.33 ± 5.00 cm	6.26 ± 4.65 cm	5.58 ± 3.07 cm	7.37 ± 4.68 cm	7.51 ± 4.90 cm	6.82 ± 3.88 cm
TWS	4.20 ± 3.93 cm	5.80 ± 2.51 cm	4.54 ± 4.08 cm	4.59 ± 3.69 cm	5.14 ± 3.89 cm	2.36 ± 1.41 cm

## Data Availability

The data supporting the reported results can be found at the following link: https://drive.google.com/drive/folders/1pS-cCW5ied2APIJ20-kFIRlKvfMiiHTO?usp=sharing (accessed on 11 June 2023).

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
