# Peer review of "Estimation of Spatio-Temporal Parameters of Gait and Posture of Visually Impaired People Using Wearable Sensors†"

_sensors, 2023, doi:10.3390/s23125564_

Round 1
Reviewer 1 Report
General comments
The topic of this study is of interest, as it focused on the development of a set of wearable inertial sensors to analyze gait and postural stability in people with visual impairments, with application in the field of biomechanics and rehabilitation.
One positive point of this work is the use of different and complementary data, which allowed for an in-depth analysis of the gait. However, the article suffers a major problem in the relationship between the aim of this study and the study design (and inferential statistics) that allows it to be achieved. Therefore, the conclusion of this study should be written more carefully. All manuscript requires an in-depth English revision.
Specific comments
Title
Why the expression "estimation" in the title? This study only had of proposing two sets of wearable inertial sensors.
Abstract
“autonomous displacement” is not a good expression for me. Review throughout the entire manuscript.
This following sentence is not clear to me: “In the current practice of rehabilitation worldwide, this assessment is carried out in person with estimates made virtually.”.
The aim of this study indicates that the spatiotemporal gait characteristics were estimated using two sets of wearable inertial sensors. How were the "real data" of the spatiotemporal features collected? What was the gold standard?
In my opinion, the gait tests should not be considered as a form of validation, considering that no inferential statistics were performed in this study that prove that the two sets of wearable inertial sensors do not present acceptable measurement errors.
Specify the study design.
How do the results of this study and the respective inferential statistics used prove that this can be a tool to be used for the assessment and re-assessment of these people?
“… walked indoor and outdoor distances at different gait velocities in their homes” seems to be a contradiction.
Keywords
The authors should choose different keywords from the title.
Order is also important, from general to specific.
Introduction
The literature review is pertinent with the topic of this study. However, the flow of ideas between paragraphs is not always the most adequate, and there is no relationship between topics (e.g.: The topic about visually impaired people seems unrelated to the previous paragraph.).
“Despite the existence of non-wearable systems designed for VIP gait analysis [1], there remains a lack of user-oriented wearable systems designed specifically for this purpose.” The authors should explain why the systems are not adapted.
“Experimental tests were conducted on 9 visually impaired volunteers, with a total of 2068 steps analyzed over 1039 meters …”. This information is not necessary in introduction.
Methods
Why is this study quasi-experimental? Was there any exposure manipulation?
- Participants:
What were the sample selection criteria?
- Procedures:
As mentioned in the abstraction section, how were obtained “the ground truth values”?
- Statistical analysis:
The descriptive statistics are not described.
Has been test the normality of variables? Which test?
Why did you choose not to perform inferential statistics?
Results
The authors presented several information of experimental procedures in this section. The results section should only be a description of the results obtained, with no place for how they were obtained or any type of interpretation. Therefore, this section needs to be rewritten.
Discussion/ conclusion
Are there limitations in this study? Which?
How strong are the study design and data (variability and statistical results) to be able to refer to what is written in the discussion/ conclusion?
All manuscript requires an in-depth English revision.
Author Response
Dear reviewer;
I am writing to thank you for your careful review of our manuscript. We appreciate your time and effort in providing us with such detailed and constructive feedback.
We have carefully considered all of your comments and have made the following changes to the manuscript:
We have arranged the introduction, including a clear description of the objective.
We have clarified the methods section and have added more detail about the statistical analysis.
We have revised the discussion section to better address your concerns, including the comparison of accuracy to current literature.
We have made minor edits to the text throughout the manuscript to improve clarity and grammar. The English was revised by an English professional.
We believe that these changes have significantly improved the manuscript. We are confident that the revised manuscript is now ready for publication.
We would like to thank you again for your helpful feedback. We appreciate your assistance in improving our manuscript.
Please follow the details of revision in the attatched document:

Reviewer 2 Report
I thank the editor for inviting me to review this study.
This work aimed to estimate gait and posture parameters of visually impaired patients, using low-cost algorithms and data from IMUs. Nine visually impaired patients were recruited and asked to walk in five different experimental conditions. Parameters of gait, which are step count (SC), step length (SL), gait velocity (GV), total displacement (D) and postural assessment (PA), were computed using methods involving one (THS) or two sensors configuration (TWS). Overall, results showed that TWS methods were more precise but that THS methods had an acceptable level of accuracy. Given that THS method has less hardware and computational cost, authors of this work suggest its use to assess gait and postures in O&M rehabilitation.
While the topic is really interesting, the manuscript lacks clarity. In the introduction, the rationale and the main objective of the work are unclear. Important sections are missing in the methods (no information about participant’s walking instruction, signal processing, data analysis, IMU’s location and characteristics, calibration and methods of synchronization). Too much methods details are presented in the results section and the discussion does not sufficiently cover the literature.
More details are presented below.
MAJOR
Introduction
l.70-85: Too much details here. The rationale and aim are unclear. Several objectives seem to be presented but what is the primary objective of this work ? Is it a study about the development of low-cost gait analysis algorithms ? Is it a study aiming to validate these new gait analysis algorithms ? Is it a study aiming to compare THS and TWS ? Is it an observational study aiming to understand how VIP walk ? This needs to be clarified in this section.
Methods
Authors should be more descriptive for materials and experimental procedure:
- How were participants instructed to walk? With/without aid? Any ‘practice’ trial before data acquisition?
- How and where (body parts) were the IMUs fixed? How were they synchronized? Were they connected with a computer?
- What was the sample frequency of the IMUs that were used? Please describe the characteristics of their components as well.
- Any information about IMUs calibration?
- Any information about signal processing?
- Any information about data analysis?
Results
Overall, the authors should consider displacing the experimental description and computation details presented in the Results section to the methods. The methods used to calculate mean errors should also be described in the methods.
Discussion
This part does not sufficiently cover the literature. For instance, several methods exist to compute parameters that were used in this study (using one, two or more sensors). I suggest authors to describe the other methods presented in the literature and compare their accuracy results with these of this work.
MINOR
Introduction
l41-46: Please cite a reference referring to IMU’s validity/reliability in motion analysis.
Methods
The authors should consider using a Figure/table to clarify the five experimental procedure and to explain where the IMU were fixed.
Results
l133-137: These sentences are redundant with the experimental procedure section of the methods. Please remove them.
l159: “More conditions where incorporated…”, please revise “More conditions were incorporated…”
l228: “N the one hand…”, please revise “On the one hand…”
Conclusion
l301-304: “The addition of a second wearable sensor increases both computational and hardware cost of TWS, which is the significant architectural difference between the gait estimation methods. Therefore, for measuring O&M rehabilitation parameters of gait the TWS method could be more appropiate.” This statement seems contradictory, I assume the authors meant that THS method could be more appropriate.
l304: Please change “appropriate” for “appropiate”.
Author Response
Dear reviewer;
I am writing to thank you for your careful review of our manuscript. We appreciate your time and effort in providing us with such detailed and constructive feedback.
We have carefully considered all of your comments and have made the following changes to the manuscript:
We have arranged the introduction, including a clear description of the objective.
We have clarified the methods section and have added more detail about the statistical analysis.
We have revised the discussion section to better address your concerns, including the comparison of accuracy to current literature.
We have made minor edits to the text throughout the manuscript to improve clarity and grammar. The English was revised by a English professional.
We believe that these changes have significantly improved the manuscript. We are confident that the revised manuscript is now ready for publication.
We would like to thank you again for your helpful feedback. We appreciate your assistance in improving our manuscript.
Please follow the details of revision in the attatched document.

Round 2
Reviewer 2 Report
I thank the authors for the changes made. I am now pleased to accept the manuscript in its current form